# Insight into Biological Control Potential of *Hirsutella citriformis* against Asian Citrus Psyllid as a Vector of Citrus Huanglongbing Disease in America

**DOI:** 10.3390/jof8060573

**Published:** 2022-05-27

**Authors:** Orquídea Pérez-González, Ricardo Gomez-Flores, Patricia Tamez-Guerra

**Affiliations:** Departamento de Microbiología e Inmunología-Laboratorio de Inmunología y Virología, Facultad de Ciencias Biológicas, Universidad Autónoma de Nuevo León, UANL, San Nicolás de los Garza, Nuevo León 66455, Mexico; operezg@uanl.edu.mx (O.P.-G.); ricardo.gomezfl@uanl.edu.mx (R.G.-F.)

**Keywords:** biocontrol agents, *Diaphorina citri*, entomopathogenic fungi, Huanglongbing (HLB) disease, insect parasitization, metabolites production

## Abstract

Studies on *Hirsutella citriformis* Speare are scarce. Among these, some reports have focused on phenotypic identification, based on its morphological structure and morphometric characteristics. This fungus is known to control economically important citrus crop pests. In recent years, *H. citriformis* has received increased attention as a control agent for the Asian citrus psyllid *Diaphorina citri* Kuwayama (Hemiptera: Liviidae), which causes the Huanglongbing (HLB) disease. Unfortunately, formal *H. citriformis* strains characterization is marginal, which mainly involves the role of biologically active exudates (metabolites) produced during their growth. Information regarding their mode of action and biocontrol potential is limited. However, epizootics reports of this fungus, under suitable environmental conditions for its development (25 °C to 28 °C and ~80% relative humidity), have demonstrated its parasitization efficacy. Therefore, it becomes challenging to determine whether *H. citriformis* strains may be developed as commercial products. In this review, we showed relevant information on isolation and bioassay strategies of *H. citriformis* to evaluate potential biocontrol strains under laboratory and field conditions in America.

## 1. Introduction

In recent years, it has become critical to find environmentally friendly and viable alternatives to reduce the negative impact of chemicals against insect pests on crops worldwide. Entomopathogenic fungi were one of the first microorganisms used for biological control of pests and about 100 genera and 700 species of fungi with entomopathogenic potential have been registered to date. Species used in mycoinsecticide formulations include *Beauveria bassiana* (Balsamo) Vuillemin, *B. brongniartii* (Sacc.) Petch, *Metarhizium anisopliae* (Metchnikoff) Sorokin, *Cordyceps confragosa* (formaly *Lecanicillium lecanii*, *Hirsutella thompsonii* Fisher, and *Cordyceps fumosorosea* Wize (formaly *Isaria fumosorosea*) [1,2]. In this regard, from more than 171 of fungi-formulated products, 33.9% includes *B. bassiana* or *M. anisopliae*, and 5.8% and 4.1% involve *C. fumosorosea* Wize (formaly *Isaria fumosorosea*) and *B. brongniartii*, respectively [3,4]. Within this extensive range of fungi genera, some are generalists such as *Beauveria* and *Metarhizium*, which infect a wide range of hosts and are the most commercially applied, whereas others are selective or specialists that only affect certain insects and are less commercially used. This is the case of *Moelleriella libera* Webber (formal *Aschersonia aleyrodis)*, which infects only the tobacco whitefly *Bemisia tabaci* (Hemiptera: Aleyrodidae), *M. rileyi* (Farlow) Samson (formerly *Nomuraea rileyi*), which infects only Lepidoptera larvae, and *H. thompsonii*, which mostly infects mites [5]. Among the specialist fungi, *H. citriformis* is a poorly studied fungus that has recently attracted attention due to its potential to cause epizootics by the Asian citrus psyllid *Diaphorina citri*, which is the vector of the Huanglongbing (HLB) disease) [6,7,8]. In this review, we conducted a literature review of the most relevant and up-to-date information on *H. citriformis*, including identification by traditional and modern methods, diverse biocontrol strategy reports, and laboratory and field studies in America.

## 2. *Hirsutella citriformis* Identification

*H. citriformis* was first described in 1920 by Speare, followed by Petch in 1923 [9], and Main in 1951 [10]. It commonly infects Hemiptera insect species, and its morphology has been previously reported [6,7,10,11,12,13,14]. This fungus is classified within the Ascomycota Division, Pezizomycotina subphylum, Sordariomycetes class (Sung et al. 2007), Hypocreomycetidae subclass, Hypocreales Order, Ophiocordycipitaceae family, *Hirsutella* genus, and *citriformis* Species. Members of this group are the anamorphs of *Cordyceps*, *Ophiocordyceps*, or *Torrubiella* [15]. 

The typical *H. citriformis* mycelium is septate, fine, and hyaline and may present synnemata and phialides (conidiogenous cells). It has a spherical base with elongated necks that bear a single conidium at the tip, which has an elongated, allantoic, cimbiform, or fusiform shape (orange segment shape), covered with an ovoid mucilaginous material (Figure 1A). Some *H. citriformis* species are slow-growing fungi and produce limited conidia [6,7,12,14].

The most comprehensive description of *H. citriformis*, which was considered a reference of morphometric characterization until 2014, was reported by Mains in 1951 [10]. To date, this is the only *Hirsutella* species reported parasitizing *D. citri* [6,7,8,12] but phialide and conidium diameters of such species [10] are different from those of *D. citri*-infecting *Hirsutella* strains in Mexico (Figure 1B).

In this regard, *H. citriformis* reproductive structure measurements directly obtained from the host [7,16,17] or reproductive structures developed in culture medium [6,12,18] are different from previous reports on these species [10] (Figure 2). In host-derived species, isolates are found in different hemipteran species and geographic regions, which may result in diverse fungal structures development. It is expected that structures dimensions developed in culture media would be similar but we observed that even isolates from the same country showed differences in their size, even if cultivated in the same culture medium. This observation was among isolates from insects parasitized by fungus collected in different months in states with different temperature and humidity conditions.

*Hirsutella citriformis* synnema dimensions on their hosts are variable, which may relate to the host insect’s size [10] (Table 1).

## 3. *Hirsutella citriformis* Origin and Distribution

Previous reports have shown that *H. citriformis* origin was in the Asian continent [9,11], with distribution to other tropical or subtropical climate geographical regions. It is possible that human activity through commercial transportation of agricultural products worldwide (mainly maritime) had played a significant role in *H. citriformis* parasitized-insects spreading to other regions, including Cuba [19], Argentina [13]), and the United States of America [7,8] (Table 2).

## 4. Insects Associated with *H. citriformis*

*Hirsutella citriformis* was first reported by Speare in 1920 [11], followed by Peath in 1923 [9]. It was further detected in the 1940s in different insect species [10]. In the 21st century, most reports of *H. citriformis* isolates were exclusively from *D. citri* insects (Table 3). It is a typically synnematous species that infects a wide range of hemipterans, including insect vectors of plant diseases. It has been detected in hemipterans in Asian countries [6,9,11], South America [13,19], the Caribbean [17,21], and the United States of America [7,8]. *H. citriformis* parasitizes several insect populations, causing epizootics under suitable environmental conditions for its development [6,7,8,21].

In Mexico, *H. citriformis* has been isolated from *D. citri* Kuwayama (Hemiptera: Liviidae) carcasses, whose first isolate was reported in Veracruz state in 2008, after which it has been detected in different locations, mainly in Mexican citrus areas [12,18,25,27,28] (Table 4).

*H. citriformis*-parasitized insects in citrus ecosystems have allowed the frequent isolation of various strains of this fungus. For this reason, in recent years, *H. citriformis* has received significant interest as a potential biocontrol agent against the Asian citrus psyllid, present in most citrus growing regions worldwide.

## 5. *H. citriformis* Mode of Action

The precise mode of action of *H. citriformis* is unknown but might be similar than that reported by Boucias et al. on *Hirsutella* genera [29]. The infection process begins after susceptible insects contact conidia, which adhere to their cuticle external surface and penetrate the hemocele, where they convert the tissues into biomass and develop mycelium inside (Figure 3).

After the insect dies, mycelium protrudes, showing *H. citriformis* characteristic structures. Its conidium binding to the insect is facilitated by a mucilaginous envelope. Culturing on artificial agar medium has demonstrated mycelial compatibility among strains, anastomosis occurrence, and one nucleus/conidium [30]. Furthermore, fungus vegetative stages have been studied in *D. citri* hemocele, where numerous hyphal bodies have been observed, showing a morphology close to the vegetative cells produced by other entomopathogenic Hypocreales fungi [29]. The *H. citriformis* mucilage that surrounds the conidium has the same protective potential than that reported for *H. satumaensis* at temperatures above or below the optimum, or after exposure to ultraviolet radiation by extended periods, cold stress, or high osmotic pressure [31]. In other fungi, the mucilage role is not clear but its contribution to host adhesion has been reported in nematodes [32].

## 6. Adverse Effects of Environmental Factors on *H. citriformis*

Although it is important to understand how environment decreases *H. citriformis* stability and viability in field, few studies on this matter have been reported. *H. citriformis* behavior and persistence under natural conditions are unknown, for which it is important to develop strategies to prevent mortality or improve its dissemination under field conditions to achieve a successful biocontrol product. In general, temperature, relative humidity (RH), and solar radiation are the most important abiotic environmental factors affecting entomopathogenic fungi germination, vegetative growth, and conidia viability [33].

### 6.1. Temperature

Temperature reduces *H. citriformis* fungus viability and affects conidium germination, development, and survival. Its optimal development temperature is 25 ± 2 °C, which is delayed and inhibited at lower and higher temperatures, respectively [34]. Pérez-González et al. [14] also reported strain-related differences based on the geographical region of isolation. They observed high-temperature resistance among strains isolated from a region that maintains elevated temperature and humidity throughout the year [14]. Furthermore, Orduño-Cruz et al. [34] showed that conidium germination was affected by the incubation temperature. In this regard, conidia have a higher germination percentage at 25 °C in a shorter incubation time, whereas at higher or lower temperatures, longer incubation time was required. They also reported that germination time varied depending on the isolate [34].

### 6.2. Relative Humidity

RH is one of the most important environmental factors altering the efficacy and survival of an entomopathogen. *Hirsutella citriformis* distribution studies under field conditions have shown that fungal infection of insects was higher in the months of high humidity and optimal temperature. High mortality of psyllid adults was reported in Réunion Island by *H. citriformis*, when the RH was higher than 88% [23]. Furthermore, maximum infection rates of 52.2% and 82.9% in *D. citri* by *H. citriformis* was observed in two Indonesian orange orchards, during September, February, and July, when the RH was higher than 80% [6]. *H. citriformis* naturally infesting *D. citri* populations was also observed at a RH higher than 80%, which was crucial for their growth and sporulation [21]. By 2016 in Colima, Mexico, an overall incidence of 58.2% of *H. citriformis* in citrus trees infested by *D. citri* was detected by PCR [28]. Similarly, *D. citri* mortality rate by *H. citriformis* was 23%, as reported in a two-year field test in Florida [8]. Authors observed up to 75% psyllid carcasses mummified and covered by synnemata, during the fall and winter months but absent during spring, probably due to a low RH [8]. Dead psyllids remained on citrus leaves during 68 days on average [8], thus indicating high fungus dissemination during template and high humidity seasons. In addition, authors did not recommend applying copper hydroxide, petroleum oil, or elemental sulfur at high rates to avoid killing this entomopathogenic fungus. In a follow-up study, authors indicated that *D. citri* adults killed by *H. citriformis* was as low as three to four percent on young leaves but six times higher than on mature leaves, suggesting the use of copper sulfate pentahydrate, aluminum tris, and alpha-keto/humic acids along with *H. citriformis* [35].

Higher infectivity of the fungus in the months in which the temperature ranged from 23 °C to 26 °C and the RH was higher than 80% was reported [26]. Moreover, *H. citriformis* presence on *D. citri* adults in Tamaulipas, Mexico was detected at temperatures between 25 °C and 28 °C and ~80% RH [24]. 

### 6.3. Solar Radiation

Several reports showed that entomopathogenic fungus conidia are inactivated in hours or days by direct sunlight [36]. However, the effect of solar radiation on *H. citriformis* conidia viability is unknown. Nevertheless, this fungus naturally adheres to mummified carcasses on the underside of citrus leaves, where they remain for more than two months and maintain infectivity to *D. citri* adults [8]. Another characteristic of this fungus is its location in the mummified carcasses, keeping in a ready-to-infect position. Healthy adults, looking to mate, may be attracted to the mummified carcass, either by visual or chemical signals [8].

## 7. Metabolites Production

Most microorganisms, including fungi, produce a wide variety of compounds or toxins (secondary metabolites) that have diverse activities. Unlike entomopathogenic fungi such as *B. bassiana* and *Metarhizium*, scarce information regarding metabolites produced by *Hirsutella* is available. It has been reported that different *Hirsutella* species, after growing in submerged fermentation, produce a variety of metabolites, some of which have activity against insects and mites (Table 5).

The best-characterized toxin is the Hirsutellin A (HTA), which is an isolated and sequenced protein produced by *H. thompsonii* [37,38]. It also produces oosporin, which reduces eggs production in mite females [40]. Hirsutellines production has been also reported by *H. nivea* [45], which possess antibacterial activity against *Mycobacterium tuberculosis*. Furthermore, exopolysaccharides from *Hirsutella* sp. have antibacterial activity against Gram-positive bacteria such as *Bacillus subtilis* and *Micrococcus tetragenus* [46]. Several benzene and phthalic acid compounds with potential insecticidal activity were detected in *H. citriformis* culture supernatants [49] (Table 5).

It has been also reported the presence of droplet exudates after growing *H. citriformis* strains on commercial agar media [50]. To date, these metabolites have not been studied to establish their toxicity against other organisms, such as insect pests. In fact, polysaccharides produced by other entomopathogenic fungi may stimulate fungus infection [51], whereas proteins promote fungus adherence, infection, and virulence [52]. We have observed that Mexican *H. citriformis* strains produce and exudate droplets, which showed at least three colors (light yellow, reddish brown, and crystalline), when grown on commercial potato dextrose agar media, supplemented with yeast extract at 1% *w*/*v* (Figure 4).

The initial metabolites analysis indicated that light yellow and reddish-brown color exudates are of proteinaceous nature, using the Bradford method [53], whereas the use of the DuBois method [53] showed that the crystalline exudates were carbohydrates [14]. *H citriformis* exudate metabolites are of mucilaginous-like consistency, showing the presence of ~60 kDa, 20 kDa, 14 kDa, and 2 kDa to 5 kDa proteins, as measured by polyacrylamide gel electrophoresis (Figure 5).

Based on these results, the molecular mass of the ~14 KDa protein might correspond to hirsutellin. We have collected and purified mucilaginous metabolites from *H. citriformis* liquid cultures, tested them under laboratory and field conditions against *D. citri* adults, and demonstrated their toxicity to *D. citri* adults (unpublished data).

## 8. *Hirsutella citriformis* Epizootics on *Diaphorina citri*

*H. citriformis* causes epizootics on *D. citri* in various citrus regions, where high humidity is a determining factor for this fungus to infect and disseminate. In this regard, psyllids control by *H. citriformis* at RH higher than 80% has been reported [54]. In Cuba, *D. citri* nymphs and adults infected with *H. citriformis* in Mexican lemon trees (*Citrus aurantifolia* Swingle) and in *Murraya paniculata* (Lin) Jack plants (limonaria) were observed [17], whereas in Mexico and other countries, detection of *H. citriformis* infecting *D. citri,* where the RH was close to or higher than 80%, was also reported. In the municipality of Tuxpan, Veracruz, Mexico, an infection rate of 21.1% of *D. citri* by *Hirsutella* sp., where the RH was from 50% to 60%, was detected [25]. In addition, *H. citriformis* infecting *D. citri* in southern Tamaulipas, Mexico (municipalities of Ciudad Victoria, Mante, Gómez Farías, and Llera) was observed, where up to 43% of insects on orange trees and 95% on *M. paniculata* were mycosed [12]. Epizootics of *H. citriformis* infecting *D. citri* in Rio Bravo, Tamaulipas, Mexico, were reported in September and November, where the RH was close to 80% and temperature averaged 26 °C [24]. Moreover, a high prevalence of *D. citri* mycosed insects by *H citriformis* in the citrus regions of Troncones, Misantla, and Iztacuaco in Tlapacoyan, Veracruz, Mexico, where the RH was higher than 80% and temperature averaged 25 °C, was observed [26].

These reports confirm that temperatures ranging from 23 °C to 28 °C and RH close or higher than 70% improves *H. citriformis* infection and dissemination on *D. citri*.

## 9. Assessments of *H. citriformis* Biocontrol against *D. citri* and *Bactericera cockerelli*

To determine the pathogenicity of *H. citriformis*, laboratory bioassays have shown the potential of *H. citriformis* conidia to infect and kill *D. citri* and *B. cockerelli* (Šulc) (Hemiptera: Triozidae) adults, which became infected after fungus contact [7,12,37,55]. It is recognized that *H. citriformis* requires longer periods to cause insect death, compared with other entomopathogenic fungi, but limited studies have evaluated the role of this fungus to infect and kill *D. citri* nymphs. Experiments where nymphs were exposed to mycosed adults by contact resulted in *H. citriformis* infection and mortality at the significant rates of 75% and 10%, respectively [7,56]. In addition, conidia application by spraying was evaluated under laboratory conditions [57], showing that adults’ mortality was lower than that following direct application of conidia. After using this spraying technique but adding an adherent to the suspension, mortality rates were closer to those achieved by this fungus after direct application. 

*H. citriformis* virulence was evaluated against *D. citri* adults, spraying selected strains in a semi-field bioassay, evidencing a mortality close to 50% and their dispersion to other citrus areas [58]. A similar study was developed in Campeche (18°50′11″ N 90°24′12″ O), Chiapas (16°24′36″ N 92°24′31″ O), Colima (19°14′37″ N 103°43′51″ O), and Quintana Roo (19°36′00″ N 87°55′00″ O) Mexico, showing that *H. citriformis* was a promising alternative to use against Hemiptera insects in integrated pest management programs [59].

In a northeast area in Montemorelos, Nuevo Leon, Mexico (25°11′14″ N 99°49′36″ O), we performed two bioassays against *D. citri* adults in September 2020 and September 2021, applying formulated conidia with two different gummous metabolites as adherents (*Acacia* sp. gum and *H. citriformis* produced gum). We demonstrated that *Hirstuella* gum control (without conidia) treatment resulted in close to 50% mortality, whereas the mortality by formulation with conidia and *Hirsutella* gum treatment was close to 80%. These results showed that *H. citriformis*-produced gum is toxic to *D. citri* (unpublished data).

## 10. Effects on Predators and Other Non-Target Arthropods

Unlike other entomopathogenic fungi such as *B. bassiana* and *M. anisopliae* that have a wide range of hosts, including 100 arthropod species and representing a potential treat to beneficial insects, *H. citriformis* has potential to infect Hemiptera insects. Nevertheless, it has only been isolated from this type of insect, representing a low-risk biocontrol entomopathogen to beneficial insects. To demonstrate *H. citriformis* safety against two of the most important predators in biological control, conidia were applied on *Hippodamia convergens* Guerin-Meneville (Coleoptera: Coccinellidae) adults and *Chrysoperla rufilabris* Burmeister (Neuroptera: Chrysopidae) nymphs, showing that treated insects were uninfected by *H. citriformis*, thus indicating that this fungus was not pathogenic against such predators [55].

## 11. Conclusions

*Hirsutella citriformis* is a fungus first described in 1908. Since then, few studies have been developed to better understand its morphological, growth, and genetic characteristics. From 2000, it has been reported as a biocontrol agent against the insect vector of the HLB disease. *D. citri* is currently considered the most important pest to citrus areas worldwide and numerous studies have been reported. Relevant *H. citriformis* characteristics include (a) slow growth and poor conidia production; (b) presence of synnemata; (c) its phialides have a spherical base, with elongated necks that have a single conidium at the tip, which has an elongated, allantoic, cimbiform, or fusiform shape, and it is covered by a mucilaginous layer; (d) it has been found parasitizing only Hemiptera class insects worldwide and appears to be safe for predatory beneficial insects; (e) it requires warm temperatures and high RH (higher than 70%) for its development; (f) it produces a variety of secondary metabolite, whose biological and chemical nature has not been yet elucidated; (g) its secondary metabolites mode of action against *D. citri* adults is unknown; (h) it causes epizootics on infected insects, under environmental conditions suitable for its development; and (i) it has shown pathogenicity against *B. cockerelli* and *D. citri*, under laboratory and field conditions. Based on pathogenicity results, we may expect that *H. citriformis* possesses potential as bioinsecticide under an inoculative-augmentative biological control scheme. However, factors that may limit the formation of epizootics should be studied.

Prevailing environmental conditions in the treated area are decisive for the establishment of a further epizootic. Therefore, areas recording long periods (at least one month) of high humidity (higher than 70%) and temperatures between 23 °C and 28 °C should be selected for *H. citriformis* application. In Mexico and other countries, citrus growing areas with these environmental conditions that allow *H. citriformis* development may be the target for its application as biological agent to reduce *D. citri* vector populations.

## Figures and Tables

**Figure 1 jof-08-00573-f001:**
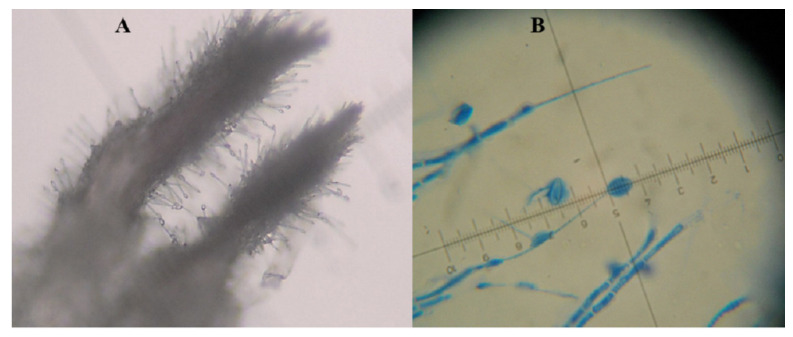
*Hirsutella citriformis* mycelium structures. (**A**) *H. citriformis* synnemata developing on *Diaphorina citri* and (**B**) phialide and conidia. Measurements are in micrometers.

**Figure 2 jof-08-00573-f002:**
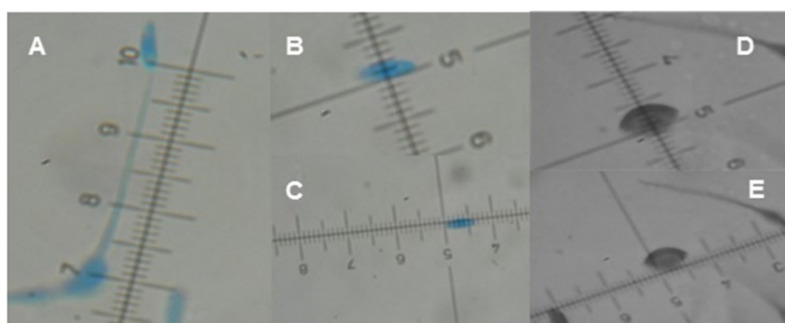
Mexican *Hirsutella* strains morphological characteristics. (**A**) Phialide length, (**B**) conidium width, (**C**) conidium length, (**D**) conidium width, including the mucilaginous envelope, and (**E**) conidium length, including the mucilaginous envelope. Measurements are in micrometers.

**Figure 3 jof-08-00573-f003:**
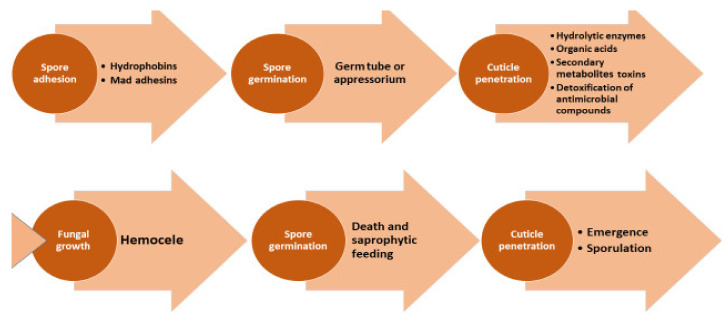
*Hirsutella thompsonii* mode of action.

**Figure 4 jof-08-00573-f004:**
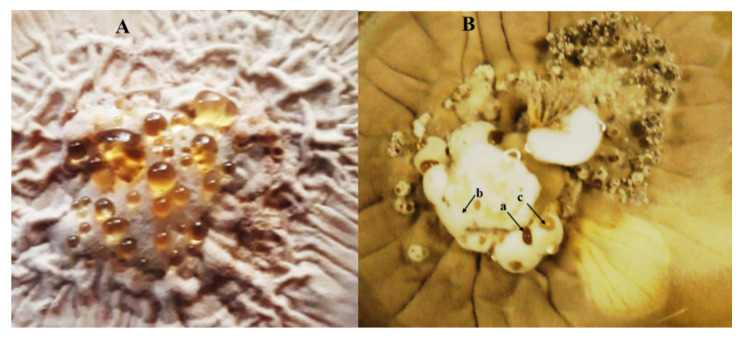
Different exudates produced by Mexican *Hirsutella citriformis* strains grown on potato dextrose agar supplemented with 0.1% yeast extract (PDAY). (**A**) *H. citriformis* strain producing mostly light brown exudates and (**B**) *H. citriformis* strain producing (a) deep brown, (b) crystalline, and (c) light brown exudates.

**Figure 5 jof-08-00573-f005:**
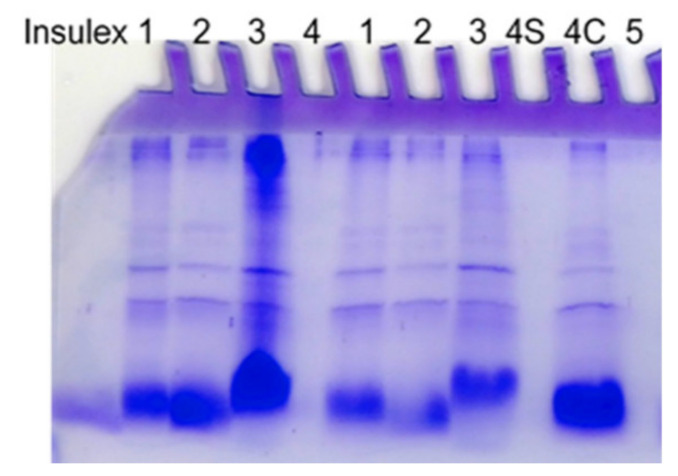
Polyacrylamide gel electrophoresis of a Mexican *Hirsutella citriformis* strain sample. Lane 1 = mycelium; lane 2 = supernatant; lane 3 = mycelium; lane 4S = gummous material dissolved in solvent and sonicated (five pulses/40 W/30 sec). This material was slightly fragmented; lane 4C = the gummous material was cut, and a fragment was directly deposited as sample; lane 5 = solvent. Note that samples dissolved in a solvent were off the gel lane.

**Table 1 jof-08-00573-t001:** Most representative reported *Hirsutella citriformis* structure measurements (µm).

Reference	Source ^1^	Phialide Length			Conidium		Mucus Diameter
		Total	Base	Sterigma	Length	Diameter	
[11]	H				5.5–8.5	1.5–1.8	
[10]	H	36.0–54.0	6.0–14.0	30.0–40.0		5.0–8.0	2.0–2.5
[16]	H	18.5–52.0				3.5–5.0	1.0–1.5
[6]	CM	27.5–62.3	5.1–9.4	22.4–52.9		6.4–7.6	2.1–2.8
[17]	H			16.8–23.6		6.8–9.1	1.5–2.3
[7]	H			17.5 ± 1.9		5.9 ± 0.8	2.6 ± 0.3
[12]	H					6.8–7.0	1.5–2.0
[13]	H	35.6–55.4		28.7–47.5		5.9–7.9	2.0–3.0
[13]	H	22.4–34.7		16.8–28.0		5.6–7.8	2.2–2.8
[14]	CM	26.0–42.0	4.0–8.5	20.0–38.0		5.4–6.3	1.6–2.0

^1^ H = host; CM = culture media.

**Table 2 jof-08-00573-t002:** *Hirsutella citriformis* geographic distribution.

Country	Region	Reference
**Cuba**	Jovellanos, Matanzas	[17]
	Carmelina in Cienfuegos; Morón and Ceballos in Ciego de Ávila; San Antonio de los Baños in La Habana	Cabrera et al., 2004, cited by [20]
	Jaguey El Grande, Matanzas	[19]
**Argentina**	Los Hornos and La Plata, Buenos Aires	[13]
**United States**	Hawaii	[11]
	Polk, Marion, and Hendry, Florida	[7]
	Indian River, Florida	[8]

**Table 3 jof-08-00573-t003:** *Hirsutella citriformis* isolation reports from different insects belonging to the Hemiptera order.

Parasitized Insect (Family)	Location	Reference
Fulgoridea family Latreille	New Zeeland	[11]
*Ricania discalis* Germar (Ricaniidae)	New Zeeland	
*Perkinsiella sacharicida* Kirkaldy (Hemíptera: Cicadellidae)	Hawaii	
*Siphanta acuta* Walker (Hemiptera: Flatidae)	Hawaii	
Pentatomidae family Leach	India	[9]
*Bothriocera venosa* Fowler (Cixiidae)	Puerto Rico	Gregory & Martorell 1940, cited by [10]
*Leptopharsa constricta* Champion (Tingidae)	United States	[10]
*Corythuca ulmi* Osborn & Drake (Tingidae)	United States	[10]
*Heteropsylla cubana* Crawford (Psyllidae)	Malaysia	[22]
*Oliarus dimidiatus* Berg (Cixiidae)	Argentina	[23]
		[13]
*Diaphorina citri* Kuwayama (Liviidae)	Rénion island, France	[17]
	Cuba	[6]
		[19]
	Indonesia	[21]
	Isla Guadalupe	[7]
	Florida	[8]
	Veracruz, MX	[24]
	Tamaulipas, MX	[12]
	Tamaulipas, MX	[25]
	Veracruz & Puebla, MX	[26]
	Veracrux, MX	[27]
	Campeche, MX	[14]
	Colima, MX	
	Chiapas, MX	
	Quintana Roo, MX	
	San Luis Potosí, MX	
	Tabasco, MX	
	Veracruz, MX	
	Yucatán, MX	
	Hidalgo, MX	Pérez-González et al., (unpublished data)
	Oaxaca, MX	

**Table 4 jof-08-00573-t004:** States and municipalities of Mexico reporting *Hirsutella citriformis*.

State	Municipality	Reference
Campeche	Edzna	[18]
Chiapas	Tapachula	[18]
Colima	Tecomán	[18]
Hidalgo	Santa Ana	Pérez-González et al., (unpublished data)
Oaxaca	Melchor Ocampo	Pérez-González et al., (unpublished data)
Puebla	La Legua and La Garita	[26]
Quintana Roo	Nuevo Israel	[18]
San Luis Potosí	Xolol	[18]
Tabasco	Huimanguillo	[18]
Tamaulipas	Rio Bravo	[24]
	Gómez Farías and Llera	[12]
Veracruz	Tlapacoyan	[18]
	Troncones, Ixtacuaco and El Lindero	[26]
	Tuxpan	[25]
Yucatán	Mococha	[18]

**Table 5 jof-08-00573-t005:** Toxins produced by *Hirsutella* species.

*Hirsutella* Species	Toxin	Type of Toxin	Toxin Size/Yield	Activity	Reference
*H. thompsonii strain* JAB-04	Hirsutellin A	Dot-blot anti HtA	16.3 kDa/0.12 mg/mL	Insecticidal vs. *Galleria mellonella*	[37,38]
*thompsonii*	Hirsutellin A Hirsutellin B	Ribotoxin	130 amino-acids	Insecticidal vs. *Galleria mellonella* Insecticidal vs. *Drosophila melanogaster*	[39]
*H. thompsonii* strain HtM120I	Hirsutellin A	Dot-blot anti HtA	16 kDa (200–400 ng/mL)	Insecticidal vs. *Tetranychus urticae*	[40]
*H. thompsonii*	Culture exudates	Phomalactones?	Undetermined	Reduces eggs production in mites	[40]
*H. rhossilensis*	Hnsp	SAAPF-pNA protease activity & others	30 kDa	Nematicide	[41]
*H. thompsonii*	Hirsutellin A	W Blot NH_2_ terminal	16 kDa	Insecticidal vs. *Aedes aegypti*, *Galleria mellonella*	[42]
*H. thompsonii*	Hirsutellin A	Ribotoxin	130 aa	Unidentified	[43]
*H. thompsonii*	Hirsutellin A	Dot-blot anti HtA	481–936 ng/mL	Insecticidal vs. *Galleria mellonella*	[44]
*H. nivea* strain BCC2594	Hirsutellones A, B, C and D	Alkaloids	29.9, 1696, 22.6 and 15.7 mg/L	Antibacterial against *Mycobacterium tuberculosis*	[45]
Unidentified sp.	Exo-polysaccharide (EPS)	Mannose (man), galactose (gal), glucose (glc)	23 kDa	Gram (+) bacteria, *Bacillus subtilis*, *Micrococcus tetragenus*	[46]
*H. beakdumountain*	EPS	EPS 1 y 2: gal, glc, man EPS3: man, glc	EPS1 = 43 kDa EPS2 = 19.5 kDa EPS3 = 4.7 kDa	Potential to eliminate hydroxyl radicals	[47]
	Intracellular polysaccharide	IPS 1 y 2: gal, glc, man IPS 3: man, glc	IPS1 = 23.1 kDa IPS2 = 21.5 kDa IPS3 = 10.4 kDa	Potential to eliminate hydroxyl radicals	[47]
*H. sinensis*	EPS	Polysaccharides (65–70%) + protein (25%)	5 kDa–200 kDa	Antioxidant activity	[48]
*H. satumaensis* Aoki	Mucilage from aerial conidia	Protein content, oligo-saccharides + man	0.12 mg/mL	Insecticidal vs. *Galleria mellonella Plutella xylostella*	[30]
*H. citriformis*	Culture supernatant	Benzene and phthalic acid	Unidentified	Insecticidal activity	[49]

## Data Availability

Not applicable.

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
