# Peer review of "Insight into Biological Control Potential of Hirsutella citriformis against Asian Citrus Psyllid as a Vector of Citrus Huanglongbing Disease in America"

_jof, 2022, doi:10.3390/jof8060573_

Round 1

Reviewer 1 Report

This manuscript is a revised version of the mini-review paper I reviewed previously (March 2022). I appreciate the authors' revision, which reads much better although it is not flawless. I also appreciate those changes the authors made in response to my previous comments and suggestions, including correction of some fungal names based on updated fungal taxonomy. Overall, the manuscript has been improved a great deal. Due to the biocontrol potential of Hirsutella citriformis against the hemipteran vector of citrus Huanglongbin disease destructive for orange production in America, I recommend the revised manuscript to be considered for acceptance after minor revision of English writing.

Suggestions:

  1. Many long phrases and sentences should be modified or simplified throughout the text based on English grammar.
  2. The first paragraph of Conclusion needs a rewriting. This long paragraph has only three periods and hence is hard to follow up.
  3. Improve citation skill. Since references are cited with digital codes for conciseness, sentences should not start by mentioning authors' names, such as Halbert and Manjunath [46] reported..., Godoy-Ceja and Cortez-Madrigal [34] detected..., Pérez-González et al. [48] applied. Try to use passive voice for description of cited story.

Author Response

We appreciate the reviewers’ effort reviewing this manuscript. Manuscript grammar was reviewed as edited. We hope that the revised version of this manuscript will be appropriate for publication in the JoF journal

Reviewer 2 Report

I agree with the changes made by the authors.

I propose Accept after minor revision, taking into account that it be reviewed by an English speaker, to help make the text excellent.

Author Response

We appreciate the reviewers’ effort reviewing this manuscript. Manuscript grammar was reviewed as edited. We hope that the revised version of this manuscript will be appropriate for publication in the JoF journal

This manuscript is a resubmission of an earlier submission. The following is a list of the peer review reports and author responses from that submission.

Round 1

Reviewer 1 Report

This review article is intended to provide information on the isolation and bioassay strategies to evaluate biocontrol potential of Hirsutella citriformis strains under laboratory and field conditions in America. This fungus is getting attraction of scientists to use as biocontrol agent against Asian citrus psyllid. Infact, the contents of the review article did not match the proposed Title – I noticed that the authors did not provide any information on the isolation and bioassay strategies using H. citriformis against citrus psyllid. The authors claimed that studies on this fungus are scarce, however, I am quoting one book chapter (https://www.sciencedirect.com/science/article/pii/B9780128234143000435) and one research article (https://escholarship.org/uc/item/548158dd) which has not been considered by the authors to include in their review. This article lacks novelty and is based mainly on the basic information on the subject – the write up and organization of the article does not commensurate with the standards of Journal of Fungi.

I am of the opinion that this review article cannot be accepted in current form for its publication – however, the Editor may allow authors to substantially improve the article and re-submit as new submission to be considered for its publication in the journal

Reviewer 2 Report

I consider it is a valuable study because Studies on Hirsutella citriformis are scarce. Besides this species is recognized to control economically important pests, affecting citrus crops.

I make the following suggestions to the authors

Line 70:  Should use “Members of this group are the anamorphs of Cordyceps, Ophiocordyceps or Torrubiella” instead of “Members of this group are the imperfect state of Cordyceps, Ophiocordyceps or Torrubiella

Line 77: (Subandiyah et al. 2000 [6, 7, 12, 14]. In the text, reference numbers should be placed in square brackets [ ], and placed before the punctuation; for example [1], [1–3] or [1,3].

Line 82 -85 Explain more exhaustively the following idea “In addition, H. citriformis reproductive structure dimensions directly obtained from the host [7, 16, 17] or reproductive structures developed in culture medium [6, 12, 18] disagree with previous reports on this species [10] (Fig. 2).”

Line 109: It is the word “corpses” appropriate?

Line 112: The phase “H. citriformis was first isolated by Speare in 1920” sound repetitive. I recommend a different wording

Line 120-137: I recommend an original figure illustrating the H. citriformis mode of action

Some parts of the text are unnecessary highlighted in gray

Line 139 – 198 I recommend to focuses in adverse effects of environmental factors on H. citriformis emphasizing in the America region. From my point of view first is important to exemplify  with American studies and then mention other studies in other latitudes. Besides I suggest to theorize how this environmental can affect the biology H. citriformis taking into account the extreme clime variability across American geography.

Line 235: Revise the relevance of the statement “Preliminary results indicate that it causes D. citri adults mortality, without requiring the fungus (unpublished data).”

Line 284-291: The paragraph corresponds to unpublished data.  I suppose the authors have obtained the data but it is not clear.

Line 306-332: I think that in the CONCLUSIONS should be more focused to America since the title is : “ Isolation and bioassay strategies of Hirsutella citriformis to control Huanglongbing (HLB) disease-causing Hemiptera in America”

Line 338 – 540 References should be revised.

All Figures, Schemes and Tables should be inserted into the main text close to their first citation and must be numbered following their number of appearance

Table 1: The term “H” and “M“ must be explained since all table columns should have an explanatory heading.

Table 2. Should emphasize the Hirsutella citriformis distribution in America

Table 5. The species of Hirsutella  in the first column should be under the 'dual taxonomy' rules of fungal nomenclature .  For example “H. thompsonii”  instead of “thompsonii”

Figure 1: I suppose Figure 1 are originals. Are this Mexican Hirsutella strains? It is necessary to put the scale bars (––– Bars = 10 µm). I recommend a better figure B). Authors should revise if the figure shows the synnemata or actually shows phialide and conidia. I think Figure B should be “B) phialide and conidium.

Figute 2: I recommend a better figure showing Mexican Hirsutella strains morphological characteristics. I recommend photomicrographs without the interference of the ocular micrometer.  ? It is necessary to put the scale bars (––– Bars = 10 µm).

Figure 3. It is necessary to put the scale. I recommend a better figure showing different coloration: a) dark brown exudate; b) crystalline exudate; c) light brown exudate.

Reviewer 3 Report

This mini-review manuscript compiles a body of information on the isolation, morphology, geographic distribution and natural prevalence of Hirsutella citriformis, which is an important mycopathogen of Asian citrus psyllid as a vector of citrus Huanglongbing disease in America. I appreciate the authors' effort to promote research interests in the development of this insect mycopathogen against the vector pest of the citrus disease and encourage them to carefully revise the manuscript. Despite limited reports on the fungus, the prepared tables are informative enough and provide a quick overview to the current status of studies on the fungus. Of course, the manuscript has a large space to be improved for clarity, conciseness and readability. I suggest the authors to consider professional editing service or a native English speaker's help for improvements of English writing and grammar.

Minor suggestions:

  1. Title change: Insight into biological control potential of Hirsutella citriformis against Asian citrus psyllid as a vector of citrus Huanglongbing disease in America
  2. The authors need pay attention to taxonomical changes of some fungal names mentioned in the manuscript, for examples:

Cordyceps fumosorosea = Isaria fumosorosea

Cordyceps confragosa = Lecanicillium lecanii

Moelleriella libera = Aschersonia aleyrodis

  1. Please pay attention to the format of JoF and keep the revised manuscript consistent with the format. It would be better to use the JoF format template for resubmission.
  2. In Table 1, what means for H and W in the column of Source?
  3. Table 5 seems too busy. Try a horizontal table to show each case study or use abbreviations of some long terms or names and give their definitions in footnotes.
  4. Figure 1. Is there any possibility to present a scale bar for the microscopic image?
  5. Figure 2. Rulers are not good scale bars for the images.